# CHRDL1 Regulates Stemness in Glioma Stem-like Cells

**DOI:** 10.3390/cells11233917

**Published:** 2022-12-03

**Authors:** Inka Berglar, Stephanie Hehlgans, Andrej Wehle, Caterina Roth, Christel Herold-Mende, Franz Rödel, Donat Kögel, Benedikt Linder

**Affiliations:** 1Experimental Neurosurgery, Department of Neurosurgery, Neuroscience Center, Goethe University Hospital, 60590 Frankfurt am Main, Germany; 2Department of Radiotherapy and Oncology, Goethe University Hospital, 60590 Frankfurt am Main, Germany; 3Division of Experimental Neurosurgery, Department of Neurosurgery, University Hospital Heidelberg, INF400, 69120 Heidelberg, Germany; 4German Cancer Consortium DKTK Partner site Frankfurt/Main, 60590 Frankfurt am Main, Germany; 5German Cancer Research Center DKFZ, 69120 Heidelberg, Germany; 6Frankfurt Cancer Institute (FCI), Theodor-Stern-Kai 7, University of Frankfurt, 60590 Frankfurt am Main, Germany

**Keywords:** glioma, glioblastoma, glioma stem-like cells, CHRDL1, BMP4

## Abstract

Glioblastoma (GBM) still presents as one of the most aggressive tumours in the brain, which despite enormous research efforts, remains incurable today. As many theories evolve around the persistent recurrence of this malignancy, the assumption of a small population of cells with a stem-like phenotype remains a key driver of its infiltrative nature. In this article, we research Chordin-like 1 (CHRDL1), a secreted protein, as a potential key regulator of the glioma stem-like cell (GSC) phenotype. It has been shown that CHRDL1 antagonizes the function of bone morphogenic protein 4 (BMP4), which induces GSC differentiation and, hence, reduces tumorigenicity. We, therefore, employed two previously described GSCs spheroid cultures and depleted them of CHRDL1 using the stable transduction of a CHRDL1-targeting shRNA. We show with in vitro cell-based assays (MTT, limiting dilution, and sphere formation assays), Western blots, irradiation procedures, and quantitative real-time PCR that the depletion of the secreted BMP4 antagonist CHRDL1 prominently decreases functional and molecular stemness traits resulting in enhanced radiation sensitivity. As a result, we postulate CHRDL1 as an enforcer of stemness in GSCs and find additional evidence that high CHRDL1 expression might also serve as a marker protein to determine BMP4 susceptibility.

## 1. Introduction

Glioblastoma (GBM) is highly aggressive and the most common brain tumour in adults. According to WHO criteria, it is classified as a grade 4 astrocytoma [1,2], and despite an intensive treatment regimen consisting of maximally possible surgery, followed by radiochemotherapy using the alkylating agent Temozolomide, the median survival barely exceeds one year [3,4,5,6]. One key characteristic of GBM is its highly infiltrative growth, which makes relapses virtually unavoidable [7]. It is hypothesised that recurrences are further supported by tumour cells that can transiently obtain a stem-like phenotype. These glioma stem-like cells (GSCs) [8,9,10] or tumour-initiating cells are thought to be able to replenish the entire tumour and are considered particularly resistant to conventional therapy, including radiotherapy and chemotherapy [11,12,13]. It has been further shown that these cells reside in specific niches, namely hypoxic or perivascular areas, and express proteins associated with stemness, whereas they lack protein expression associated with differentiated neuronal cells [12,14,15,16,17,18].

Recently, we demonstrated that arsenic trioxide (ATO) efficiently induces the differentiation of GSCs and inhibits proliferation while inducing cell death, particularly in combination with the natural anticancer component (-)-Gossypol (Gos, also known as AT-101) [19]. The proteomic analysis further revealed that multiple processes associated with DNA repair, but also associated with stemness, were effectively depleted. Among these proteins, we identified Chordin-like 1 (CHRDL1) as a potential key regulator of the GSC phenotype. CHRDL1 is a secreted protein that antagonises the function of bone morphogenic protein 4 (BMP4) by binding to its receptors [20]. BMP4, in turn, has been shown to induce GSC differentiation and reduce tumourigenicity upon the transplantation of BMP4-treated GBM cells into nude mice [21]. Additionally, it has been proposed that BMP4 treatment of GSCs induces asymmetric divisions favoring stem-like daughter cells [22], while Sachdeva et al. proposed that BMP signaling mediated by BMP4-treatment induces quiescence [23]. Notably, BMP proteins are antagonised by secreted proteins of the Chordin family [24], including CHRDL1 [25]. In particular, Cyr-Depauw et al. showed that CHRDL1 inhibits the migration and invasion of breast cancer cells via CHRDL1-mediated BMP4 inhibition [25]. Interestingly, Gao et al. employed rat neural stem cells and demonstrated that CHRDL1 induces neuronal differentiation [26]. We, therefore, hypothesised that CHRDL1 acts as a GSC-derived inhibitor of BMP4 and thereby maintains GSC stemness. In this study, we show that the depletion of the secreted BMP4 antagonist prominently decreases functional and molecular stemness traits, resulting in enhanced radiation sensitivity.

## 2. Materials and Methods

### 2.1. Cells and Cell Culture

The experiments were conducted with the following glioma stem-like (GSC) lines: NCH644 [27] and GS-5 [28]. All cell lines were maintained in a neurobasal medium (Gibco, Darmstadt, Germany). The medium was supplemented with 1 × B27, 100 U/mL Penicillin 100 µg/mL Streptomycin (P/S, Gibco), 1 × GlutaMAX (Gibco), 20 ng/mL epidermal growth factor (EGF, Peprotech, Hamburg, Germany) and 20 ng/mL fibroblast growth factor (FGF, Peprotech) and 2 µg/µL or 0.5 µg/µL Puromycin (Santa Cruz Biotechnology Inc., Heidelberg, Germany), respectively. The GSCs were dissociated using Accutase (Sigma-Aldrich, Taufkirchen, Germany) to create a single-cell suspension prior to seeding. All cells were tested monthly for mycoplasma using the PCR Mycoplasma Test Kit II (AppliChem, Darmstadt, Germany) according to the manufacturer’s instructions. Christel Herold-Mende (University Hospital Heidelberg, Germany) provided NCH644, and GS-5 was gifted by Katrin Lamszus (UKE, Hamburg, Germany). Both lines were generated from surgical specimens using the mechanical and enzymatic dissociation and cultivation of free-floating spheroids as described [27,28]. HEK293T cells (ATCC, Manassas, VA, USA #CRL-3216) were cultured in Dulbecco’s modified Eagle’s medium (DMEM GlutaMAX) and was supplied with heat-inactivated 10% FBS and P/S (all from Gibco).

### 2.2. Lentiviral Transduction of GSCs

The lentiviral transduction of shRNA targeting CHRDL1 (shCHRDL1) or mammalian non-targeting Control-shRNA (shCtrl) was performed as described previously [29,30] after the transfection of the shRNA-containing vector (pLKO.1-puro, Sigma-Aldrich), gag/pol-plasmid (psPAX2, addgene, Watertown, MMA, USA #12260) and VSV g envelope plasmid (pMD2.G, addgene #12259) into HEK293T cells. Viral supernatants were collected after 16 h and additionally after 24 h before pooling and mixing with a fresh medium (ratio: 1:1) supplied with 8 µg/mL protamine sulfate (Sigma-Aldrich) immediately prior to transduction. To select for positively transduced cells, 2 µg/mL and 0.5 µg/mL puromycin were added and maintained in the culture 48 h post-transduction for NCH644 and GS-5, respectively. pLKO.1-puro plasmids containing the shCHRDL1 or shCtrl (SHC002)-sequences were purchased from Sigma-Aldrich. To deplete CHRDL1, one of the following sequences was used:

3′-CCGGGAGAACTGTCATGGGAACATTCTCGAGAATGTTCCCATGACAGTTCTCTTTTTTG-5′(TRCN0000149739) and

3′-CCGGACGCCATGCACAGCATAATTTCTCGAGAAATTATGCTGTGCATGGCGTTTTTTG-5‘(TRCN0000371790) for NCH644 and GS-5, respectively.

### 2.3. Cell-Based Assays 

#### 2.3.1. MTT (3-(4,5-Dimethylthiazol-2-yl)-2,5-Diphenyltetrazolium Bromide) Assay 

The MTT assay was performed as described previously [31] and measured on a Tecan Spark (Tecan, Männedorf, Switzerland) plate reader at 560 nm. MTT (Sigma-Aldrich) was solved in sterile PBS at 5 mg/mL. For NCH644, 8000 cells per well were seeded for 5-time points in a 96-well plate.

#### 2.3.2. Limiting Dilution Assay

The limiting dilution assay (LDA) was performed as described previously [19,32]. To summarise, 96-well plates were used to seed the cells in a 200 µL culture medium per well. By performing a row-wise descending dilution, cell concentrations of 8, 16, 32, 64, 128, 256, 512, and 1024 cells/well for NCH644 and 16, 32, 64, 128, 256, 512, 1024, and 2048 cells/well for GS-5 were reached. Stem-cell frequencies were assessed 7 days (NCH644) and 7 and 14 days (GS-5) after seeding using extreme limiting dilution analysis (ELDA) software which employed the standard settings (http://bioinf.wehi.edu.au/software/elda; [33]; last accessed on 5 September 2022). 

#### 2.3.3. Sphere Formation Assays

To measure the sphere-forming ability, an assay (SFA) based on Gilbert et al. [34] was performed. Briefly, 500 and 1000 dissociated NCH644 and GS-5, respectively, were seeded in 96 well plates and measured after 7 days. Images were acquired using a Tecan Spark plate reader and analysed using a self-developed macro via FIJI (v1.52p) [35], which enhanced contrasts and identified sphere area and properties as described previously [29]. 

#### 2.3.4. Cell Cycle Analysis

A total of 150,000 cells were seeded in a 6-well plate for 24 h. For analysis, the medium was removed, the cells were washed with PBS, and a single cell suspension was prepared using Accutase-incubation for 10 min at 37 °C. The cells were next harvested in FACS tubes and were washed with 100 µL cold PBS. An amount of 3 mL of ice-cold 70% ethanol was added dropwise to the cells by simultaneous vortexing, followed by incubation for 2 h on ice. The FACS tubes were centrifuged for 3 min at 1000 rpm to form pellets. For analysis, cells were incubated for 5 min at room temperature with 50 µL RNAse A (Qiagen, 20 µg/mL) and followed by 150 µL Propidium Iodide (Sigma-Aldrich, 50 µg/mL) for 30 min at room temperature. The flow cytometric determination of the cell cycle was performed by counting 20,000 cells on an Accuri C6 (Becton Dickinson, Franklin Lakes, NJ, USA) operated through BD Accuri C6 software (Version: 1.0.264.21; Becton Dickinson) and by measuring no more than 500 events/s.

### 2.4. SDS-PAGE and Western Blot

Western Blotting was carried out as described previously [36]. After blocking with 5% bovine serum albumin (Carl Roth GmbH, Karlsruhe, Germany) (BSA)/Tris-buffered saline (TBS)-Tween 20 (TBS-T) or 5% dry milk (Carl Roth GmbH)/TBS-T, the primary antibodies were incubated overnight in 5% BSA/TBS-T at 4 °C, while secondary goat anti-mouse, goat anti-rabbit or donkey anti-goat antibodies (dilution 1:10,000, LI-COR Biosciences, Bad Homburg, Germany) were incubated at room temperature (RT) for 1 h. Detection and quantification were achieved using an LI-COR Odyssey reader (LI-COR Biosciences).

The following primary antibodies and dilutions were used: CHRDL1 (#AF1808, R&D Systems, Wiesbaden-Nordenstadt, Germany) 1:2000; GAPDH (#CB1001, Calbiochem, Darmstadt, Germany) 1:20,000; OLIG2 (#AF2418, R&D Biosystems) 1:5000; SOX2 (#MAB2018, R&D Systems) 1:5000; SOX9 (#ab185966, Abcam, Cambridge, UK); SMAD1 [(#6944, Cell Signaling Technology (CST), Danvers, MA, USA)] 1:1000; SMAD5 [(#12534, CST)] 1:1000; pSMAD1/5 [(#9516, CST)] 1:1000.

The following secondary antibodies were used at a dilution of 1:10,000 (all from LI-COR Biosciences): IRDye 800CW goat-anti rabbit 1:10,000 (926-32211), IRDye 680RD goat-anti mouse 1:10,000 (926-68070), and IRDye 800CW donkey anti-goat 1:10,000 (926-32214).

The quantification of Western Blot images was performed using the raw files with Image Studio Lite (Version 5.2, LI-COR Biosciences) by manual selection of the regions of interest. The signal values of the target proteins were first normalised to the housekeeping proteins and afterward to the respective control condition, which was set to 1.

### 2.5. Irradiation Procedures 

In total, 500 cells/well were seeded into 96-well plates and irradiated the next day. Irradiation (IR) with single doses of 2, 4, or 6 Gy was performed using a linear accelerator with 6 MV photon energy, a 100 cm focus to the isocentre distance, and a dose rate of 6 Gy/min (Synergy, Elekta, Crawley, UK) at the Department of Radiation and Oncology (University Hospital Frankfurt, Frankfurt, Germany). 

### 2.6. Immunofluorescence Microscopy/Foci Assay

For DNA damage analyses, 12,000 NCH644 cells/well were seeded on Laminin-coated 8-well chamber slides (Falcon, Corning, Amsterdam, NY, USA). Laminin-coating (10 µg/mL, Sigma-Aldrich, L2020) was performed at 4 °C overnight. One day after seeding, the cells were irradiated as indicated and, after an additional 24 h, were fixed with 4% paraformaldehyde for 20 min at RT. The slides were washed with TBS-Tween (0.1%; TBS-T), blocked with 4% BSA in TBS with 0.3% Triton X-100 for 1 h at RT, and the primary antibody incubation occurred at 4 °C overnight. Hereafter, the slides were washed at least three times with TBS-T, and a secondary antibody was diluted 1:500 in TBS-T and incubated for 1 h at RT. After an additional wash step with TBS-T, the slides were mounted with DAPI containing Immunoselect antifading mounting medium (Dianova, Hamburg, Germany) or Fluoroshield with DAPI (Thermo Fisher, Frankfurt, Germany). Images were acquired with an Eclipse TS100 inverted fluorescence microscope (Nikon, Düsseldorf, Germany) operated by NIS Elements AR software (version 3.22, Nikon).

The following primary and secondary antibodies were used: 53BP1 (NP-100-304, Bio-Techne GmbH, Wiesbaden, Germany), 1:1000; Alexa Fluor^®^ 488 F(ab’)2 fragment goat anti-rabbit IgG (H + L) (Thermo Fisher); 1:500. 

### 2.7. Taqman-Based qRT-PCR

In total, 300,000 cells/well (NCH644) and 600,000 cells/well (GS-5) were seeded into 6-well plates, and cells were collected the following day. Experiments were performed using 3 biological replicates for each treatment condition, while the experiment was repeated three times. RNA was isolated using the ExtractMe Total RNA Kit (Blirt S.A., Gdanks, Poland), and 1–2 µg RNA was used for cDNA synthesis. SuperScript III System (Life Technologies, Darmstadt, Germany) allowed the synthesis of cDNA, with 100 U per sample to be sufficient. The quantitative real-time PCR (qRT-PCR) was performed using Taqman-probes (Applied Biosystems, Darmstadt, Germany) and the Fast-Start Universal Probe Master Mix (Roche) on a StepOne Plus System (Applied Biosystems) in a 20 µL reaction volume. 

Ct values were normalised to the TATA box-binding protein (TBP). Fold-change in the gene expression was determined by the 2^−∆∆Ct^ method.

The following Taqman-probes were used: CHRDL1 (Hs01035484_m1), OLIG2 (Hs00300164_s1), SOX2 (Hs01053049_s1), SOX9 (Hs00165814_m1), ZEB1 (Hs01566408_m1), ZEB2 (Hs00207691_m1), CD44 (Hs01075864_m1), NES (Hs04187831_g1), GFAP (Hs00909233_m1), NEFL (Hs00196245_m1), MAP2 (Hs00258900_m1), RBFOX3 (Hs01370654_m1) and TBP (Hs00427620_m1).

### 2.8. Interrogation of the Human Protein Atlas

The human protein atlas (proteinatlas.org; last accessed on: 10 October 2022) [37] was accessed using a web browser, and the expression of CHRDL1 was analyzed among multiple cancer entities. For the generation of Kaplan–Meier plots, the “best expression cut-off” was chosen, and the reported *p*-value (log-rank-test) is presented in each sub-figure. The following datasets were used: ENSG00000101938-CHRDL1/pathology/glioma; https://www.proteinatlas.org/ENSG00000101938-CHRDL1/pathology/renal+cancer; https://www.proteinatlas.org/ENSG00000101938-CHRDL1/pathology/urothelial+cancer.

### 2.9. Statistics

Statistical analyses involved one-way and two-way ANOVA using GraphPad Prism 7 (GraphPad Software, La Jolla, CA, USA), with the respective post hoc test, as indicated. For LDA, the statistical evaluation was taken from ELDA software, which calculated the statistical significance based on a chi-square test [33]. Significances were marked as follows: *p* < 0.05: *, *p* < 0.01: **, *p* < 0.001: ***, *p* < 0.0001: ****, ns: not significant. 

## 3. Results

### 3.1. CHRDL1 Depletion Reduces Stemness of GSCs

To analyze the role of CHRDL1 in maintaining the GSC phenotype, we first established stable CHRDL1-knockdown (KD) cell lines (shCHRDL1) by lentiviral transduction with two different shRNAs directed against CRHDL1. We could show for both NCH644 [27] (Figure 1A) and GS-5 [28] (Figure 1A) that a successful KD could be achieved. Pooled data from the quantification of several independent experiments revealed that for both NCH644 and GS-5 (Figure 1B), significant depletion was achieved. Based on our hypothesis that CHRDL1 maintains stemness, we next investigated the sphere-forming potential of both GSCs via LDAs. For NCH644 shCtrl (Figure 1C), we observed a stem-cell frequency of 1/21.7, whereas, upon CHRDL1 depletion, this was reduced 15-fold to 1/333.7. Representative pictures for NCH644 and GS-5 are depicted in Figure 1D. Since GS-5 GSCs grow considerably slower, we further analyzed these cells after 1 week (Figure 1E) for better comparison with NCH644 and to allow for sufficient sphere formation after an additional week (Figure 1F) of incubation. One week after seeding, GS-5 shCtrl revealed a calculated stem-cell frequency of 1/404, whereas shCHRDL1 GS-5 had a 1.5-fold lower frequency of 1/600. This finding was further underscored after 2 weeks, which showed a stem-cell frequency of 1/75.5 of shCtrl GSCs, while shCHRDL1-GSCs had a 3.3-fold lower frequency of 1/248.7. Overall, we could show that CHRDL1 depletion reduced stemness.

An analysis of the cell cycle after CHRDL1-depletion (Figure 2) revealed that NCH644 (Figure 2A) displayed an increase in cells in the G1-phase with a concomitant decrease in S- and G2/M-phases, indicative of G1-arrest due to CHRDL1-depletion. GS-5 cells display similar cell cycle phase distribution irrespective of CHRDL1 depletion (Figure 2B), which might reflect the overall lower effect size observed using the limited dilution assay. In summary, the inhibition of proliferation might be partially accountable for the observed reduction in the stem-cell-frequency but does not fully explain our results.

Based on these findings, we therefore asked if this functional reduction in stemness was also associated with the reduced expression of stemness-associated genes. For this purpose, we performed Taqman-based qRT-PCR in both NCH644 and GS-5. For NCH644 (Figure 3), we could validate the robust depletion of *CHRDL1* mRNA expression upon shRNA-mediated CHRDL-KD (Figure 3A). Similarly, we observed a prominent reduction in the stemness marker genes *OLIG2* (Figure 3B) and *SOX2* (Figure 3C), indicating reduced stemness. Curiously, *SOX9* expression (Figure 3D) shows a slight but significant induction after CHRDL1 depletion. We further analyzed *ZEB1* (Figure 3E) and *ZEB2* (Figure 3F), which are associated with enhanced stemness, but also with an EMT-like phenotype, and we observed moderate yet significant repression (*ZEB1*) and induction (*ZEB2*), respectively. The transmembrane protein *CD44* (Figure 3G) has also been associated with the invasiveness of GBM cells and is considered a stemness marker, and this gene was strongly reduced upon CHRDL1-KD, while the intermediate filament NES (Nestin) is markedly and strongly depleted (Figure 3H). Finally, we analyzed a set of differentiation-associated genes. Upon CHRDL1 depletion, the glial marker *GFAP* (Figure 3I), as well as the neuronal marker *NEFL* (Neurofilament, Figure 3J), were significantly upregulated, whereas *MAP2* (Figure 3K) only displayed a tendency. Lastly, *RBFOX3* (Figure 3L), another neuronal marker, was significantly increased. Collectively these data indicate that CHRDL1-depletion potently blocks several pathways associated with stemness and invasion while inducing a more differentiated cellular state.

Similar to NCH644, we analyzed the same gene panel for GS-5 GSCs (Figure 4) and first validated CHRDL1 depletion (Figure 4A). Furthermore, we confirmed the efficient depletion of the stemness-associated genes *OLIG2* (Figure 4B) and *SOX2* (Figure 4C), which is also accompanied by a significant reduction in *SOX9* expression (Figure 3D). The EMT-associated gene *ZEB1* (Figure 4E) is also significantly decreased, while *ZEB2* (Figure 4F) and *CD44* (Figure 4G) remain unchanged. The intermediate filament gene *NES* (Figure 4H) only displays a slight but significant reduction after CHRDL1 depletion. Of the four differentiation markers, *GFAP* (Figure 4I), *NEFL* (Figure 4J), *MAP2* (Figure 4K), and *RBFOX3* (Figure 4L) neither show a significant change, although *GFAP* and *MAP2* show a slight tendency towards increased expression. In summary, we could validate our findings of grossly reduced stemness in a second GSC line, further underscoring that CHRDL1 could be a novel master regulator of GBM stemness.

Using GS-5 GSCs, we also validated the loss of the protein expression for the stemness-associated proteins OLIG2, SOX9, and SOX2 via Western Blot (Figure 5) and observed a robust depletion of these proteins.

### 3.2. CHRDL1 Depletion Re-Activates BMP4-Signaling in GSCs

Based on the notion that a high CHRDL1 expression blocks BMP4 signaling and thereby enforces an enhanced stem-like state of GSCs, we first confirmed that NCH644 is proficient at relaying BMP4-mediated signals by treating the cells with the recombinant human BMP4 ligand (Figure 6A). This showed us that upon BMP4 treatment, pSMAD1/5 was highly increased, indicative of active BMP4-signaling. Similarly, CHRDL1-depleted NCH644 GSCs (Figure 6B) also displayed an increased pSMAD1/5 expression. The treatment of NCH644 and GS-5 GSCs with recombinant BMP4 (Figure 6C,D) resulted in a reduced sphere formation potential in both GSCs, as did CHRDL1 depletion. The treatment of CHRDL1-depleted cells with rBMP4 led to no further enhancement in NCH644 compared to the solvent-treated shCHRDL1 GSCs, whereas for GS-5, the combined approach further and potentially synergistically reduced the sphere-forming potential.

### 3.3. CHRDL1 Depletion Sensitizes GSCs towards Radiation Treatment

Next, we reasoned that CHRDL1-depleted cells might be more vulnerable to conventional treatment, such as radiotherapy. For this purpose, we irradiated the cells with increased doses of X-ray (single doses) and analysed the amount of DNA double-strand breaks (DNA-DSB) using immunofluorescent staining against 53BP1 (Figure 7A). We observed significantly more 53BP1-positive foci, indicative of enhanced DNA damage, in CHRDL1-depleted cells, which were mock irradiated and after irradiation with 2 Gy. After 4 Gy treatment, no significant differences were observed, whereas irradiation with 6 Gy significantly increased the amount of 53BP1 foci compared to non-irradiated GSCs. Hereafter, we reasoned that higher amounts of DNA-DSBs might lead to enhanced growth inhibition and therefore performed sphere formation assays of freshly dissociated GSCs seeded as single cells (Figure 7B). This approach revealed that in CHRDL1-expressing NCH644 shCtrl cells, every IR dosage significantly impedes sphere formation. CHRDL1 depletion without irradiation (0 Gy) also significantly impedes sphere formation, basically confirming our results shown above. The combination of irradiation and CHRDL1 depletion can further block sphere formation after treatment with 6 Gy, thus, reflecting the enhanced amount of DNA-DSBs. Representative pictures are depicted in Figure 7C. Hereafter, we sought to confirm that this decrease was due to the re-activation of BMP4-signaling and performed a similar experiment using NCH644 cells treated with recombinant BMP4 (Figure 7D), which could confirm the result obtained with the CHRDL1 shRNA. Hence, we observed that BMP4 treatment reduces sphere sizes, similar to irradiation. The combination of both treatments significantly further reduced sphere areas, indicative of a profound sensitization via BMP4-signaling to the irradiation treatment.

### 3.4. CHRDL1 Is Associated with Poor Prognosis in Glioma and Other Cancers

Lastly, we interrogated the human protein atlas [37] in order to unravel the relevance of CHRDL1 for patient outcomes and observed that high CHRDL1 expression was associated with worse clinical outcomes in glioma (Figure 8A), while it could also be considered unfavourable in urothelial (Figure 8B) and renal cancer (Figure 8C).

## 4. Discussion

Glioblastoma remains one of the most dismal cancer diagnoses in adults, and current treatment strategies fail to cure the patients. It is hypothesized that this is largely due to stem-like states that can be obtained by some or all residual tumour cells that survive surgery and radiochemotherapy, resulting in more aggressive and ultimately lethal tumours. A better understanding of the underlying mechanisms is crucial to develop targeted and patient-centred approaches that can be based on the molecular knowledge of different tumour types. 

BMP4 was shown to enforce the differentiation of GSCs almost two decades ago and has been tested as a “differentiation therapy” [21]. Although this notion has not been fulfilled and the entire concept of stable differentiation has been challenged, particularly due to the presence of different, highly plastic cell states that might be driven by different molecular alterations [8,9,10], the question remains as to how tumour cells evade these stemness-blocking cues. One factor that has been shown recently to be overexpressed in cancers, including GBM, is the secreted BMP4 antagonist CHRDL1 [24,25]. Accordingly, our working hypothesis was that CHRDL1 acts as an enforcer of stemness in GSCs. Within this report, we employed two previously described GSC spheroid cultures [27,28] and depleted them of CHRDL1 using the stable transduction of a CHRDL1-targeting shRNA. This CHRDL1 depletion was accompanied by functional as well as molecular changes that conclusively can be interpreted as stemness blockade. Furthermore, considering that GSCs are known to be more resistant towards conventional therapy, such as radiotherapy, we could further confirm that CHRDL1-depleted cells are indeed sensitized to IR treatment. Interestingly, it has been proposed that IR treatment induces stemness in cancer cells [38,39] via the upregulation of SOX2 [38], among other known stemness factors. SOX2 is one of the major factors regulated by CHRDL1, as shown by us and others [40]. Therefore, it seems likely that CHRDL1 depletion not only counteracts this IR-induced stemness increase but even effectively prevents the stemness traits of GSCs. Similarly, BMP4 treatment would be expected to result in a similar cooperative effect, as has been confirmed by us to our knowledge for the first time in this report. Similarly, it has already been shown that the BMP4 treatment of GBM cells sensitizes them to the current gold-standard chemotherapy with Temozolomide [41], as well as to more targeted agents such as Bevacizumab [42,43].

Very recently, it was shown in diffuse intrinsic pontine glioma, a very aggressive juvenile glioma type, that the activation of BMP signalling enforces cellular differentiation, similar to our findings presented in the current study. Mechanistically it was shown that the upregulation of CHRDL1 expression results in the counteraction of the tumour-suppressive effects of BMP4 and that the depletion of CHRDL1 significantly reduces cell proliferation and sphere formation as well as tumour growth in a xenograft model [40]. The authors also showed that the BMP4 treatment of diffuse intrinsic pontine glioma (DIPG) cells induced a stemness blockade accompanied by decreased SOX2 and OLIG2 expression, further validating our findings. Mechanistically the authors proposed that by the epigenetic upregulation of the tumour suppressor CXXC5, the BMP4-response could be mediated [40]. Considering these studies, one may propose that the (re-) activation of BMP4 signalling might be a suitable strategy to block the stem-like phenotype of adult and juvenile gliomas and thereby sensitize them towards conventional therapy regimes. In our study, we found additional evidence that high CHRDL1 expression might serve as a marker protein to determine BMP4 susceptibility. Whether this mechanism might also be expanded to other tumour entities should also be investigated in future studies. For example, it is known that BMP4 can exert pro-tumourigenic functions in breast cancer, where it mediates migration and invasion, which can be blocked via CHRDL1 [25], whereas CHRDL1 induces the neuronal differentiation of neural stem cells [26]. Accordingly, the origin and/or location of the tumour likely dictates how external cues are perceived. Additionally, an open question that remains unanswered for now is whether different GBM subtypes, such as proneural, mesenchymal, and classical GBM [44], respond differently to BMP4 treatment/CHRDL1-blockade. We could show that the proneural [45] NCH644 responds moderately to BMP4 treatment while GS-5, which has been classified as a mixture of proneural and mesenchymal [28], display an additional stemness blockade by combined BMP4 treatment and CHRDL1 depletion. However, based on our very restricted sample size, it is difficult to reach a conclusive statement, and the response of different subtypes should be analysed more systematically in future studies.

Curiously, a recent report by Sachdeva et al. [23] took an opposing view from the literature and our recent findings by claiming that BMP4 treatment rather induces quiescence instead of blocking stemness. Similarly, as discussed above, this could be explained by different subtypes reacting differently to this kind of treatment. Moreover, the report by Sachdeva et al. does not necessarily contradict our findings. In their report, they employed a short-term high-dose treatment with BMP4, whereas we analysed cells stably depleted of an upstream regulator of endogenous BMP4. Hence, one plausible hypothesis is that (re-)activating BMP4 signalling first induces a shift towards quiescence/cell cycle arrest, which results in the loss of stemness traits after the continuous activation of BMP4 signalling. Whether this hypothesis remains true and whether the inhibition of BMP4-signaling re-establishes stem-like traits should be answered in future studies. In addition, it cannot be excluded that CHRDL1 also has BMP-independent functions. 

Clinically it is interesting to note that CHRDL1 is, according to data from the human protein atlas, associated with worse survival in glioma, as well as in urothelial and renal cancer. One could therefore hypothesize that CHRDL1 has the potential as a prognostic factor and could be considered in routine diagnostics. According to the literature, CHRDL1 is associated with a worse prognosis in oral squamous cell carcinoma [46], where it also regulates metastasis and EMT [47].

Interestingly, astrocytes are also able to secrete CHRDL1 during the physiological process of wound healing after ischemic injury to facilitate the healing process [48]. This leaves the open questions on where CHRDL1 originates in a multicellular tumour and how it affects non-tumour cells. Using more complex model systems, such as an organotypic brain slice culture or even in vivo approaches, these questions could be addressed in future studies, and the possibility of targeting CHRDL1 in a cell-autonomous way, e.g., via targeting antibodies, should be explored.

## Figures and Tables

**Figure 1 cells-11-03917-f001:**
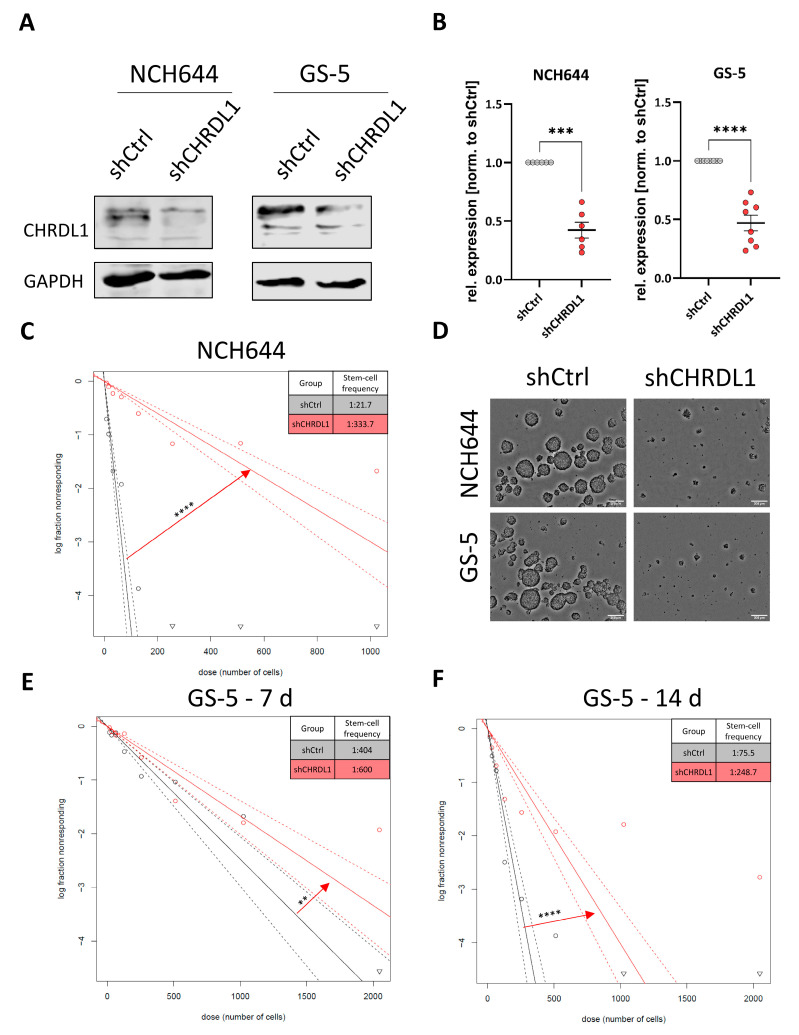
CHRDL1 depletion blocks stemness in GSCs. (**A**) Western Blot of NCH644 and GS-5 GSCs after stable transduction with a non-targeting shRNA (shCtrl) or specific shRNA targeting CHRDL1 (shCHRDL1) and (**B**) Quantification of several independent experiments. (**C**,**E**,**F**) Log-fraction plots of the limited dilution model of data from (**C**) NCH644 shCtrl (black) and NCH644 shCHRDL1 (red) GSCs 7 days after seeding the cell in a dilution series from 1024 to 8 cells and analyzing the data using ELDA software [33]. (**D**) Representative microphotographs of NCH644 and GS-5 with and without CRHDL1-depletion. Scale bar: 200 µm. (**E**,**F**) Data from GS-5 GSCs (**E**) 7 days and (**F**) 14 days after seeding of 2048 to 16 cells. **: *p* < 0.01, ***: *p* < 0.001 ****: *p* < 0.0001.

**Figure 2 cells-11-03917-f002:**
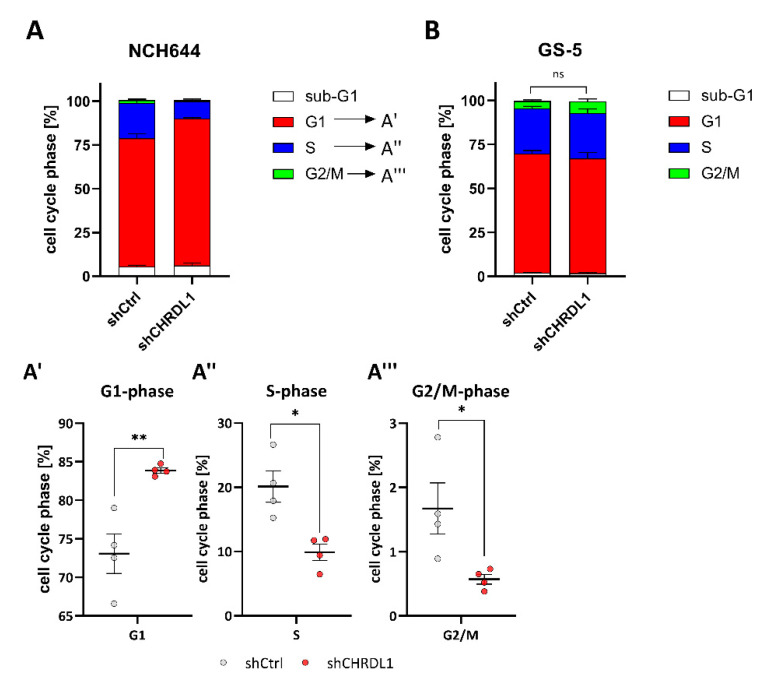
Cell cycle distribution of GSCs after CHRDL1-depletion. (**A**,**B**) Stacked bar graph of (**A**) NCH644 and (**B**) GS-5 after stable transduction with a non-targeting shRNA (shCtrl) or specific shRNA targeting CHRDL1 (shCHRDL1). After 24 h of seeding the cells, the cell cycle distribution was analyzed. For NCH644 GSCs a significant increase in cells in (**A**′) G1-phase with a concomitant significant decrease in (**A**″) S- and (**A**‴) G2/M-phases was observed. The experiment was performed using biological quadruplicates. * *p* < 0.05; ** *p* < 0.01; ns: not significant; unpaired *t* test.

**Figure 3 cells-11-03917-f003:**
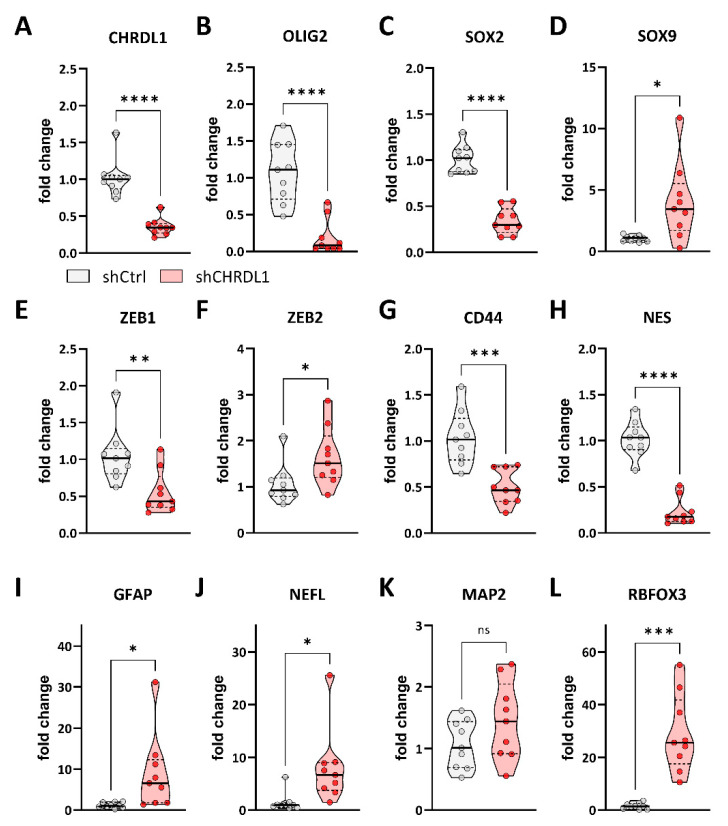
Point-plots of Taqman-based gene expression of NCH644 GSCs after stable transduction with a non-targeting shRNA (shCtrl) or specific shRNA targeting CHRDL1 (shCHRDL1) and measurement of (**A**) *CHRDL1*, (**B**) *OLIG2*, (**C**) *SOX2*, (**D**) *SOX9*, (**E**) *ZEB1*, (**F**) *ZEB2*, (**G**) *CD44*, (**H**) *NES*, (**I**) *GFAP*, (**J**) *NEFL*, (**K**) *MAP2* and (**L**) *RBFOX3* expression. The data are the summary of three experiments performed in triplicates. * *p* < 0.05; ** *p* < 0.01; *** *p* < 0.001; **** *p* < 0.0001; ns: not significant; unpaired *t* test with Welch’s correction; white dot: shCtrl, red dot: shCHRDL1.

**Figure 4 cells-11-03917-f004:**
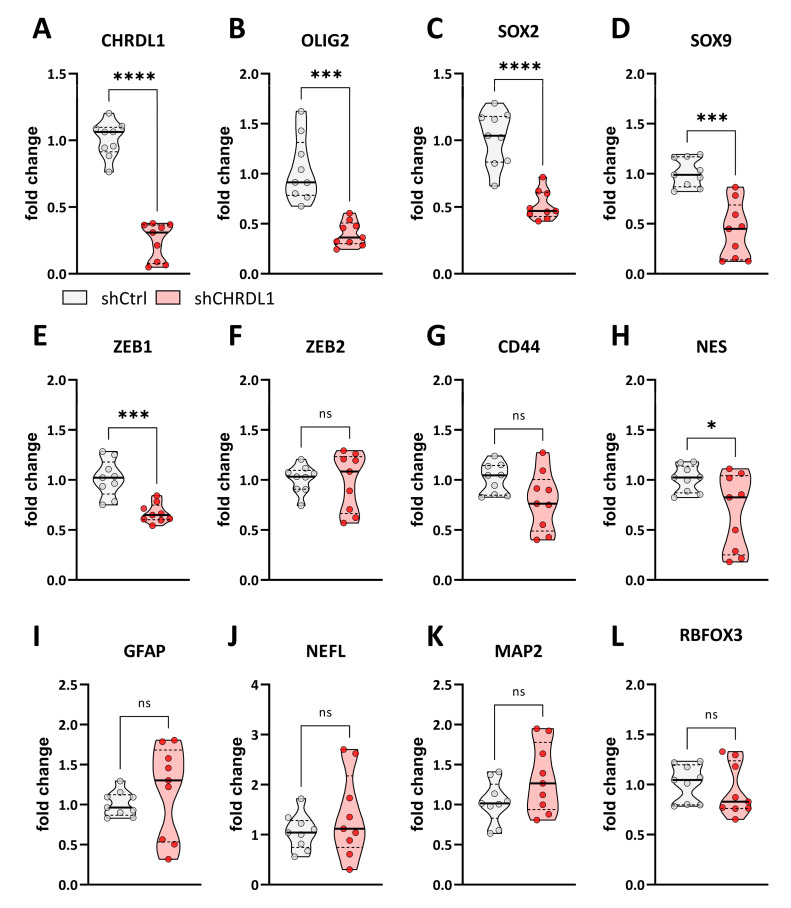
Point-plots of Taqman-based gene expression of GS-5 GSCs after stable transduction with a non-targeting shRNA (shCtrl) or specific shRNA targeting CHRDL1 (shCHRDL1) and measurement of (**A**) *CHRDL1*, (**B**) *OLIG2*, (**C**) *SOX2*, (**D**) *SOX9*, (**E**) *ZEB1*, (**F**) *ZEB2*, (**G**) *CD44*, (**H**) *NES*, (**I**) *GFAP*, (**J**) *NEFL*, (**K**) *MAP2* and (**L**) *RBFOX3* expression. The data are the summary of three experiments performed in triplicates. * *p* < 0.05; *** *p* < 0.001; **** *p* < 0.0001; ns: not significant; unpaired *t* test with Welch’s correction; white dot: shCtrl, red dot: shCHRDL1.

**Figure 5 cells-11-03917-f005:**
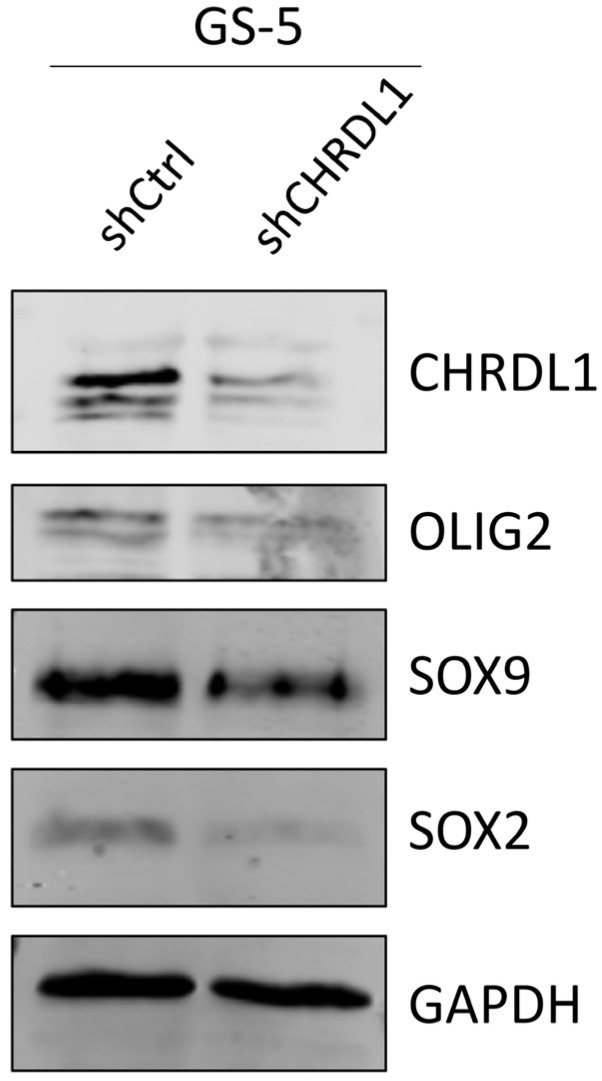
CHRDL1-depletion reduces stemness-associated marker protein expression. Western Blots of GS-5 GSCs after stable transduction with a non-targeting shRNA (shCtrl) or specific shRNA targeting CHRDL1 (shCHRDL1) and measurement of the protein expression of OLIG2, SOX9, and SOX2. CHRDL1 was detected to validate efficient depletion and GAPDH served as a housekeeping protein.

**Figure 6 cells-11-03917-f006:**
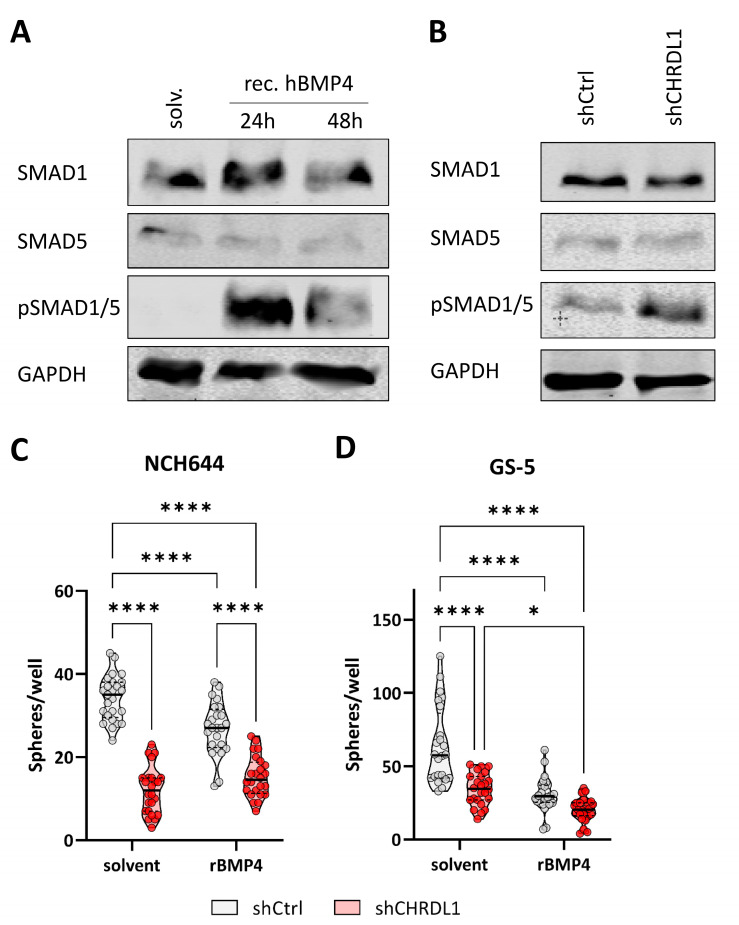
CHRDL1-depletion activates BMP4-signaling. (**A**,**B**) Western Blot of (**A**) NCH644 treated with 20 ng/mL recombinant human BMP4 for 24 or 48 h or (**B**) NCH644 shCtrl and shCHRDL1. (**C**,**D**) Sphere formation assay of (**C**) NCH644 and (**D**) GS-5 shCtrl or shCHRDL1 GSCs and treatment with 25 ng/mL recombinant BMP4 or solvent for 7 days after seeding of 500 and 1000 cells per well. * *p* < 0.05; **** *p* < 0.0001; two-way-ANOVA with Sidak’s multiple comparison tests; white dot: shCtrl, red dot: shCHRDL1.

**Figure 7 cells-11-03917-f007:**
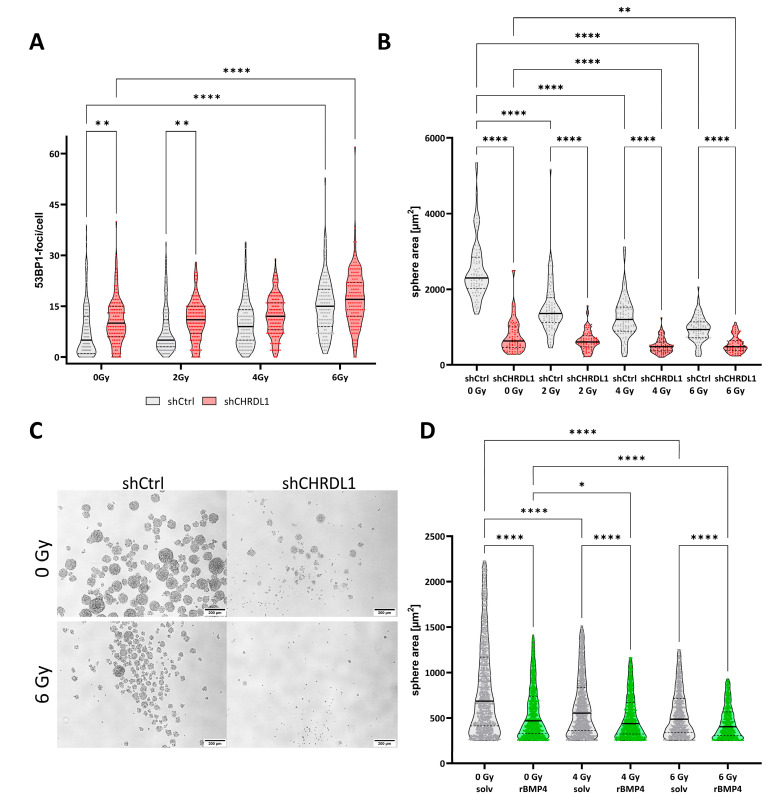
CHRDL1-depletion radiosensitizes GSCs. (**A**) 53BP1 foci assay 24 h after irradiation of NCH644 shCtrl or NCH644 shCHRDL1 GSCs with the indicated doses. (**B**) Sphere formation of NCH644 shCtrl or shCHRDL1 cells 7 days after seeding of single cells and irradiation as indicated. (**C**) Representative microphotographs of NCH644 shCtrl or NCH644 shCRHDL1 without irradiation (0 Gy) or after irradiation with 6 Gy; scale bar: 200 µm. * *p* < 0.05; ** *p* < 0.01; **** *p* < 0.0001; two-way-ANOVA with Sidak’s multiple comparison tests. (**D**) Sphere formation of NCH644 treated with solvent or recombinant human BMP4 (25 ng/mL) immediately prior to irradiation with a dose of 4 or 6 Gy. Sphere area was determined 7 days after treatment. * *p* < 0.05; **** *p* < 0.0001; Kruskal—Wallis test with Dunn’s multiple comparisons test.

**Figure 8 cells-11-03917-f008:**
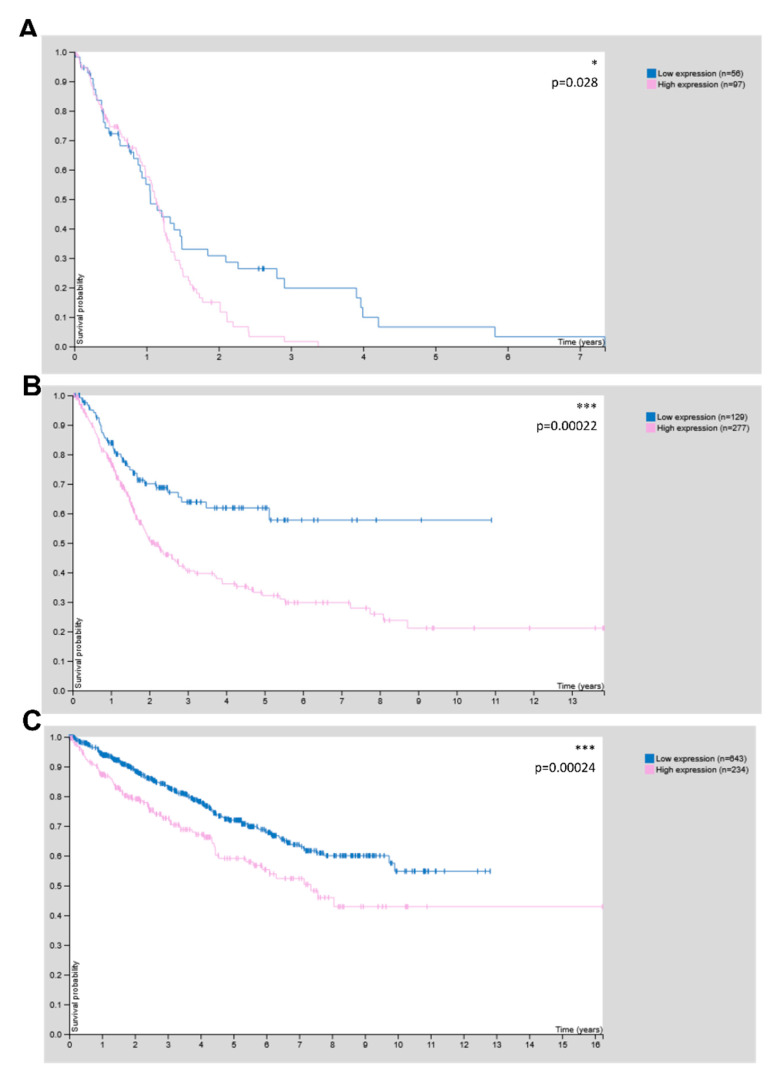
CHRDL1 is associated with worse survival in glioma and (**A**) An unfavourable prognosis in (**B**) urothelial and (**C**) renal cancer. According to data from the human protein atlas [37] (proteinatlas.org). * *p* < 0.05; *** *p* < 0.001; log-rank *p*-value for Kaplan–Meier plot. Image Credit: Human Protein Atlas.

## Data Availability

No large-scale datasets have been generated. Raw data of the experiments and/or materials can be provided upon reasonable request.

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
