# Peer review of "CHRDL1 Regulates Stemness in Glioma Stem-like Cells"

_cells, 2022, doi:10.3390/cells11233917_

Round 1

Reviewer 1 Report

In this article, the authors highlighted the importance of cancer stem cells as drivers for the infiltrative properties of glioblastoma, for the high probability of tumor relapses and for the resistance to radiotherapy. Specifically, they focused on the protein CHRDL1, which antagonizes a protein (BMP4) that is associated with GSCs differentiation, thus representing a potential key regulator of glioblastoma cells stemness. Using two GSCs cultures (NCH644 and GS-5) depleted for CHRDL1 through a lentiviral transduction of shRNA, the authors found that in absence of CHRDL1 there is a decrease in functional and molecular stemness traits: firstly, with a limiting dilution assay, they demonstrated that the two GSCs lines depleted for CHRDL1 reduced their ability in sphere formation, and then these depleted lines were assayed for the quantification of stemness- and differentiation-associated genes through a qRT-PCR, demonstrating an overall reduction of stemness markers together with an increase of differentiation markers. To further study the interaction between CHRDL1 and BMP4, the authors firstly demonstrated that the two GSCs lines can activate the BMP4-mediated pathway by adding the BMP4 ligand to the cells and assaying the presence of two proteins (pSMAD1/5) phosphorylated upon the BMP4-mediated signaling with a Western Blot, both in presence and in depletion of CHRDL1. Then, normal and CHRDL1-depleted NCH64 and GS5 cells were treated with recombinant BMP4, confirming a reduction in sphere formation upon BMP4 addition and a potential synergistic effect with CHRDL1 depletion. Then, they tested the radiosensitivity of CHRDL1-depleted GSCs by analyzing the amount of DNA Double Strand Breaks formation upon X-ray treatment: they observed that a reduced level of stemness is associated with an increased sensitivity of the tumor to radiotherapy.

The paper is well written and the experiments are well carried. This is an interesting work pointed out an identification of novel key regulators of stemness in cancer cells that might be targeted and use in combination with radiotherapy to potentially increase the rate of success of the common therapy.

I have, however, few major points that I am confident, if the authors will address them, will improve the quality of the manuscript as well as the relevance of the story.

Major:

1)      The validation of the system used by the authors must be clearly validated. Statistics for the reduction effect induced by the shRNA targeting CHRDL1 (Fig. 1A) must be displayed. If the reduction in the protein is not statistically significant, I suggest to replicate, at least for the main experiment a double shRNA to better reduce the CHRDL1 protein level.

2)      The change in stemness markers can have a dramatic influence in the resting membrane potential of GBM cells (Yang and Brackenbury 2013, Frontiers in physiology) as well as their cell cycle I suggest to perform a cytofluorimetric analysis of GB cells cell cycle at the two main conditions (control vs CHRDL1 silencing).

3)      The mRNA analysis in Fig2 is absolutely convincing, however it is not always the case that mRNA content and corresponding protein levels are proportionally linked: some WB validation for some representative of these markers are highly encouraged as a proof of principle.

4)      Are the authors expecting an increase in the stemness markers in the experiment of Fig 5 in the control condition? I suggest to implement the experiment with the radiosensitivity using also the rBMP4

Minor:

1)      Representative picture of the different experimental conditions are always highly appreciate by the reader and gives and experimental reference to the statistics reported in the manuscript (fig 1B,C,D for example)

2)      The limitation of the experiment in Fig. 4 is that consistent results were observed only for one line (GS-5), and not for the other (NCH64), so the potentiation of stemness reduction by BMP4 treatment in CHRDL1-depleted cells should be further investigated. Can it be that in the two cell lines different pathways are involved? Or that the subtypes of cancer stem cells have a different proportions? The authors should discuss that in the conclusion section.

Author Response

In this article, the authors highlighted the importance of cancer stem cells as drivers for the infiltrative properties of glioblastoma, for the high probability of tumor relapses and for the resistance to radiotherapy. Specifically, they focused on the protein CHRDL1, which antagonizes a protein (BMP4) that is associated with GSCs differentiation, thus representing a potential key regulator of glioblastoma cells stemness. Using two GSCs cultures (NCH644 and GS-5) depleted for CHRDL1 through a lentiviral transduction of shRNA, the authors found that in absence of CHRDL1 there is a decrease in functional and molecular stemness traits: firstly, with a limiting dilution assay, they demonstrated that the two GSCs lines depleted for CHRDL1 reduced their ability in sphere formation, and then these depleted lines were assayed for the quantification of stemness- and differentiation-associated genes through a qRT-PCR, demonstrating an overall reduction of stemness markers together with an increase of differentiation markers. To further study the interaction between CHRDL1 and BMP4, the authors firstly demonstrated that the two GSCs lines can activate the BMP4-mediated pathway by adding the BMP4 ligand to the cells and assaying the presence of two proteins (pSMAD1/5) phosphorylated upon the BMP4-mediated signaling with a Western Blot, both in presence and in depletion of CHRDL1. Then, normal and CHRDL1-depleted NCH64 and GS5 cells were treated with recombinant BMP4, confirming a reduction in sphere formation upon BMP4 addition and a potential synergistic effect with CHRDL1 depletion. Then, they tested the radiosensitivity of CHRDL1-depleted GSCs by analyzing the amount of DNA Double Strand Breaks formation upon X-ray treatment: they observed that a reduced level of stemness is associated with an increased sensitivity of the tumor to radiotherapy.

The paper is well written and the experiments are well carried. This is an interesting work pointed out an identification of novel key regulators of stemness in cancer cells that might be targeted and use in combination with radiotherapy to potentially increase the rate of success of the common therapy.

I have, however, few major points that I am confident, if the authors will address them, will improve the quality of the manuscript as well as the relevance of the story.

Answer: We would like to thank the reviewer for appreciating our work and we are confident that the revised version of our manuscript has been significantly improved.

Major:

  • The validation of the system used by the authors must be clearly validated. Statistics for the reduction effect induced by the shRNA targeting CHRDL1 (Fig. 1A) must be displayed. If the reduction in the protein is not statistically significant, I suggest to replicate, at least for the main experiment a double shRNA to better reduce the CHRDL1 protein level.

Answer: Thank you very much for this remark. We agree that validation is key to robust experiments. In fact, we have frequently re-checked the presence of efficient knockdowns in our cells throughout the experiment. Accordingly, we now present pooled quantification of several independent experiments (new Fig. 1B) alongside the representative Western Blot already shown in Fig. 1A.

  • The change in stemness markers can have a dramatic influence in the resting membrane potential of GBM cells (Yang and Brackenbury 2013, Frontiers in physiology) as well as their cell cycle I suggest to perform a cytofluorimetric analysis of GB cells cell cycle at the two main conditions (control vs CHRDL1 silencing).

Answer: This is an excellent suggestion. Based on our data shown in Fig. 1 a reduction of GSCs cell cycle progression seems very likely, particularly considering the available data on BMP4-treatment of GBM cells that leads to cell cycle arrest/exit. Following the reviewers suggestion, we have now performed a cell cycle analysis at the two main conditions and can show that NCH644, which also display a stronger response to CHRDL1-depletion, can be considered as G1-arrested, while GS-5 show no difference. Obviously, a PI-staining is not capable to detect cells leaving the cell cycle (Quiescence, Senesence) and can also not be used to derive insights into the respective length of the cell cycle phases. This interesting new thoughts can be addressed in follow-up-studies.

  • The mRNA analysis in Fig2 is absolutely convincing, however it is not always the case that mRNA content and corresponding protein levels are proportionally linked: some WB validation for some representative of these markers are highly encouraged as a proof of principle.

Answer: Another excellent suggestion. We have validated selected proteins and provide these novel data as new Fig. 5 in the revised manuscript.

  • Are the authors expecting an increase in the stemness markers in the experiment of Fig 5 in the control condition? I suggest to implement the experiment with the radiosensitivity using also the rBMP4

Answer: This is an interesting question. According to the literature IR treatment is indeed described to induce stemness [1,2]and we have addressed this in the revised version. However, considering tumor heterogeneity we cannot rule out that irradiation rather enriches for resistant (or less-responsive) cells within the larger population, which could be (mis-)interpreted as an apparent stemness induction. As for BMP4-treatment, we agree that this is a valid control, and performed a sphere formation experiment using NCH644 cells treated with recombinant human BMP4 immediately prior to irradiation. This approach basically confirms the effects obtained after CHRL1-depletion and we present this new data now alongside the previous findings in new Figure 7. We have additionally expanded our discussion section about the role of BMP4 in treatment sensitization.

Minor:

  • Representative picture of the different experimental conditions are always highly appreciate by the reader and gives and experimental reference to the statistics reported in the manuscript (fig 1B,C,D for example)

Answer: An excellent suggestion. We have now added representative microphotographs to several figures.

  • The limitation of the experiment in Fig. 4 is that consistent results were observed only for one line (GS-5), and not for the other (NCH64), so the potentiation of stemness reduction by BMP4 treatment in CHRDL1-depleted cells should be further investigated. Can it be that in the two cell lines different pathways are involved? Or that the subtypes of cancer stem cells have a different proportions? The authors should discuss that in the conclusion section.

Anwer: We would like to thank the reviewer for this excellent question. Different subtypes could very well explain these differences between the two cell models, but with our current knowledge, we cannot rule out other possibilities. NCH644 have been clearly defined as proneural [3], whereas to our understanding the subtyping for GS-5 isn’t completely clear as they appear to be grouped in both proneural and mesenchymal [4]. We know from initial experiments that GS-5 express higher amounts of CHRDL1 compared to NCH644 and therefore the relative knockdown effect (comparing NCH644 and GS-5) is stronger, thus explaining why GS-5 show additive effects between CHRDL1-depletion and BMP4-treatment. But we agree that a systematic analysis among the various subtypes will provide valuable insights and have discussed this perspective now in the manuscript.

  1. Ghisolfi, L.; Keates, A.C.; Hu, X.; Lee, D.K.; Li, C.J. Ionizing radiation induces stemness in cancer cells. PLoS One 2012, 7, e43628, doi:10.1371/journal.pone.0043628.
  2. Park, H.R.; Choi, Y.J.; Kim, J.Y.; Kim, I.G.; Jung, U. Repeated Irradiation with gamma-Ray Induces Cancer Stemness through TGF-beta-DLX2 Signaling in the A549 Human Lung Cancer Cell Line. Int J Mol Sci 2021, 22, doi:10.3390/ijms22084284.
  3. Podergajs, N.; Motaln, H.; Rajcevic, U.; Verbovsek, U.; Korsic, M.; Obad, N.; Espedal, H.; Vittori, M.; Herold-Mende, C.; Miletic, H., et al. Transmembrane protein CD9 is glioblastoma biomarker, relevant for maintenance of glioblastoma stem cells. Oncotarget 2016, 7, 593-609, doi:10.18632/oncotarget.5477.
  4. Gunther, H.S.; Schmidt, N.O.; Phillips, H.S.; Kemming, D.; Kharbanda, S.; Soriano, R.; Modrusan, Z.; Meissner, H.; Westphal, M.; Lamszus, K. Glioblastoma-derived stem cell-enriched cultures form distinct subgroups according to molecular and phenotypic criteria. Oncogene 2008, 27, 2897-2909, doi:10.1038/sj.onc.1210949.

Reviewer 2 Report

In this manuscript, Berglar and collaborators studied the important role of CHRDL1 in glioma stem-like cell phenotypes. They showed the knockdown of CHRDL1 in NCH644 and GS-5 could block the stemness in GSC and induce differentiation. BMP4-mediated signaling is activated in CHRDL1 depletion cells and enforces cellular differentiation. They have identified CHRDL1 depleted cells are more sensitive to X-ray-induced DNA damage according to 53BP1 foci assay. Overall, the experiments are well-designed and executed, and the findings are interesting and novel. The manuscript is good for Cells. However, before the manuscript can be published, the authors should address the following issues.

1). The authors identified CHRDL1 is associated with worse survival in glioma perhaps by activating the BMP signaling. Previous studies have shown that BMP signaling directs astroglial differentiation in GSC is consistent with this report. I wonder if the CHRDL1 knockdown has any effect on GSC proliferation.

2). CHRDL1-depletion sensitized GSCs towards radiation treatment. When the cells are treated with BMP4, are they more susceptible to irradiation?

3). R Sachdeva et al. (2019 scientific reports, DOI 10.1038/s41598-019-51270-1) reported BMP and TGF-b signaling define divergent molecular identities in glioblastoma, quiescent and proliferative GSCs, respectively, using scRNA-seq data. Does CHRDL1 show significantly different expression levels in these two populations? Moreover, they found BMP in cell stemness, tumorigenicity and resistance are opposite to the previous findings and current study, the authors should mention it in the discussion.

Author Response

In this manuscript, Berglar and collaborators studied the important role of CHRDL1 in glioma stem-like cell phenotypes. They showed the knockdown of CHRDL1 in NCH644 and GS-5 could block the stemness in GSC and induce differentiation. BMP4-mediated signaling is activated in CHRDL1 depletion cells and enforces cellular differentiation. They have identified CHRDL1 depleted cells are more sensitive to X-ray-induced DNA damage according to 53BP1 foci assay. Overall, the experiments are well-designed and executed, and the findings are interesting and novel. The manuscript is good for Cells. However, before the manuscript can be published, the authors should address the following issues.

1). The authors identified CHRDL1 is associated with worse survival in glioma perhaps by activating the BMP signaling. Previous studies have shown that BMP signaling directs astroglial differentiation in GSC is consistent with this report. I wonder if the CHRDL1 knockdown has any effect on GSC proliferation.

Answer: We would like to thank the reviewer for the appreciation of our work and for the helpful comments that will further improve our manuscript. Considering our data presented in Fig. 1 and the fact that BMP4 has been shown to lead to cell cycle arrest/exit of GBM cells, changes in proliferation are certainly occurring and (partially) explain our findings since cells forced out of the cell cycle will stop proliferating and hence are also unable to form spheres. Our interpretation is that CHRDL1-depletion increases the amount of cell cycle-inactive cells by removing the necessary driving cues, such as the various stemness markers. To completely discern proliferation effects from stemness blockade is probably a very cumbersome and time-consuming undertaking, however we can provide additional data analysing the cell cycle distribution of CHRDL1-depleted versus control cells. We have now performed a cell cycle analysis and can show that NCH644, which also display a stronger response to CHRDL1-depletion, can be considered as G1-arrested, while GS-5 show no difference. Obviously, a PI-staining is not capable to detect cells leaving the cell cycle (Quiescence, Senesence) and can also not be used to derive insights into the length of the cell cycle phases. This interesting new thoughts can be addressed in follow-up-studies.

2). CHRDL1-depletion sensitized GSCs towards radiation treatment. When the cells are treated with BMP4, are they more susceptible to irradiation?

Answer: This excellent question was also considered by the other reviewer. According to the literature IR treatment is indeed described to induce stemness [1,2] and we have addressed this in the now-revised version. However, considering tumor heterogeneity we cannot rule out that irradiation rather enriches for resistant (or less-responsive) cells within the larger population, which could be (mis-)interpreted as an apparent stemness induction.

As for BMP4-treatment, we agree that this is a valid control, and performed a sphere formation experiment using NCH644 cells treated with recombinant human BMP4 immediately prior to irradiation. This approach basically confirms the effects obtained after CHRL1-depletion and we present this new data now alongside the previous findings in new Figure 7. We have additionally expanded our discussion section about the role of BMP4 in treatment sensitization.

3). R Sachdeva et al. (2019 scientific reports, DOI 10.1038/s41598-019-51270-1) reported BMP and TGF-b signaling define divergent molecular identities in glioblastoma, quiescent and proliferative GSCs, respectively, using scRNA-seq data. Does CHRDL1 show significantly different expression levels in these two populations? Moreover, they found BMP in cell stemness, tumorigenicity and resistance are opposite to the previous findings and current study, the authors should mention it in the discussion.

Anwer: This is an interesting idea. According to our understanding Sachdeva et al re-analyzed an existing dataset from Patel et al. [5]. Unfortunately, we have not been able to recruit a bioinformatitian during the brief revision-period and therefore have not been able to replicate the analyses from Sachdeva et al. Instead, we employed the R2 database (R2: Genomics Analysis and Visualization Platform (http://r2.amc.nl http://r2platform.com)) and firstly checked for CHRDL1 expression. Unfortunately, CHRDL1 is not detected in this dataset. Therefore, this question remains unanswered for now. As for the second part of the question, we believe first-of-all that a short-term treatment will elicit a different response compared to a stable genetic depletion. Hence our experimental approaches can only be compared to some extent. Sachdeva et al. are proposing phase-transition due to short-term pulses of BMP4-treatment directing the cells towards quiescence or rather reduced cell cycle proliferation. In contrast to that, our approach targets an upstream regulator of BMP4 using a stable depletion with some extent of remaining expression. Thus, this approach might actually result in a more physiologically balanced expression of CHRDL1 in the tumor cells and due to the stable nature any kind of induced phase-transitions might already have been occurred. Following this line of thought one plausible hypothesis is that after transitioning towards quiescence, continuous activation of BMP4-signaling might lead to the effects observed by us (blockade of stemness). In addition, we cannot rule out that CHRDL1 regulates BMP4-independent oncogenic signaling pathways that functionally further enhance re-activation of BMP4-signaling. This can be addressed in future studies. We have now further discussed this in our manuscript.

  1. Ghisolfi, L.; Keates, A.C.; Hu, X.; Lee, D.K.; Li, C.J. Ionizing radiation induces stemness in cancer cells. PLoS One 2012, 7, e43628, doi:10.1371/journal.pone.0043628.
  2. Park, H.R.; Choi, Y.J.; Kim, J.Y.; Kim, I.G.; Jung, U. Repeated Irradiation with gamma-Ray Induces Cancer Stemness through TGF-beta-DLX2 Signaling in the A549 Human Lung Cancer Cell Line. Int J Mol Sci 2021, 22, doi:10.3390/ijms22084284.
  3. Podergajs, N.; Motaln, H.; Rajcevic, U.; Verbovsek, U.; Korsic, M.; Obad, N.; Espedal, H.; Vittori, M.; Herold-Mende, C.; Miletic, H., et al. Transmembrane protein CD9 is glioblastoma biomarker, relevant for maintenance of glioblastoma stem cells. Oncotarget 2016, 7, 593-609, doi:10.18632/oncotarget.5477.
  4. Gunther, H.S.; Schmidt, N.O.; Phillips, H.S.; Kemming, D.; Kharbanda, S.; Soriano, R.; Modrusan, Z.; Meissner, H.; Westphal, M.; Lamszus, K. Glioblastoma-derived stem cell-enriched cultures form distinct subgroups according to molecular and phenotypic criteria. Oncogene 2008, 27, 2897-2909, doi:10.1038/sj.onc.1210949.
  5. Patel, A.P.; Tirosh, I.; Trombetta, J.J.; Shalek, A.K.; Gillespie, S.M.; Wakimoto, H.; Cahill, D.P.; Nahed, B.V.; Curry, W.T.; Martuza, R.L., et al. Single-cell RNA-seq highlights intratumoral heterogeneity in primary glioblastoma. Science 2014, 344, 1396-1401, doi:10.1126/science.1254257.

Round 2

Reviewer 1 Report

The authors have significantly improved the manuscript answering in an extensive way my suggestions. 

The paper has significantly improved its impact and the additional controls have increased the robustness of the results. 

For this reason I recommend this manuscript for publication. 

Kind regards,

Reviewer 2 Report

The Authors have addressed all of my concerns with the original manuscript. The revised manuscript is ready for publication.